# CAN YOUR TRUST YOUR EXPERIMENTS?
# GENERALIZABILITY OF EXPERIMENTAL STUDIES

## ABSTRACT

Experimental studies are a cornerstone of Machine Learning (ML) research. A common and often implicit assumption is that the study's results will generalize beyond the study itself, e.g., to new data. That is, repeating the same study under different conditions will likely yield similar results. Existing frameworks to measure generalizability, borrowed from the casual inference literature, cannot capture the complexity of the results and the goals of an ML study. The problem of measuring generalizability in the more general ML setting is thus still open, also due to the lack of a mathematical formalization of experimental studies. In this paper, we propose such a formalization, use it to develop a framework to quantify generalizability, and propose an instantiation based on rankings and the Maximum Mean Discrepancy. We show how this latter offers insights into the desirable number of experiments for a study. Finally, we investigate the generalizability of two recently published experimental studies.

## 1 INTRODUCTION

Experimental studies are a cornerstone of Machine Learning (ML) research. Due to their importance, the community advocates for high methodological standards when performing, evaluating, and sharing studies (Hothorn et al., 2005; Huppler, 2009; Montgomery, 2017).

The quality of an experimental study depends on multiple aspects. First, the experimenter should properly define the *scope* and the *goals* of the study. Particular attention must be given to the choice of benchmarked methods and experimental conditions (Boulesteix et al., 2015; Bouthillier et al., 2021; Dehghani et al., 2021). Second, the study should be *reproducible* by independent parties and hence contain the necessary documentation. This aspect has recently drawn much attention due to the so-called reproducibility crisis (Baker, 2016; Gundersen et al., 2023; Peng, 2011; Raff, 2023; 2021). Third, the results of the study should be sensibly analyzed to draw conclusions regarding, for instance, the *significance* of the findings (Benavoli et al., 2017; Corani et al., 2017; Demsar, 2006). Finally, the *generalizability* of a study concerns how well its results are replicated under unseen experimental conditions, such as datasets not considered in the study (National Academies of Science, 2019; Findley et al., 2021; Pineau et al., 2021). The latter two conditions are also known as the internal and external validity of a study.

Generalizability and significance, although sometimes confused, are two independent aspects of a study (Findley et al., 2021). On the one hand, significant findings may not be replicated under other conditions; on the other hand, results might consistently be not significant. Generalizability is, conceptually, closely related to model replicability. A model is $\rho$-replicable if, given i.i.d. samples from the same data distribution, the trained models are the same with probability $1 - \rho$ (Impagliazzo et al., 2022). An experimental study is generalizable if, when performed under different i.i.d. samples of experimental conditions, the results are similar with high probability (National Academies of Science, 2019). A quantifiable notion of generalizability thus requires a formalization of experimental studies, of their results, and of similarity between results.

Significance, instead, captures how strong the findings are *within* the specific sample of experiments performed. Multiple publications have shown how different choices of experimental conditions can lead to very different results (Benavoli et al., 2017; Boulesteix et al., 2017; Bouthillier et al., 2021; Dehghani et al., 2021; Gundersen et al., 2022; Mechelen et al., 2023). Some recent experimental studies have also reported this phenomenon. Matteucci et al. (2023) discuss how previous studies

on categorical encoders disagree on the best-performing ones, even when the results are significant. Similarly, Lu et al. (2023) re-evaluated coreset learning methods and found that all of the methods they considered did not beat a naïve baseline.

Quantifying generalizability can also help determine the appropriate size of experimental studies. If one dataset is probably not enough to draw generalizable conclusions, $10^6$ datasets likely are. Of course, such large studies are usually not practical: it is crucial to determine the minimum amount of data needed to achieve generalizability. This principle also applies to other experimental factors, such as the choice of quality metric and the initialization seed.

Our contributions are the following:

1. we formalize experimental studies and their results;
2. we propose a quantifiable definition of the generalizability of experimental studies;
3. we develop an algorithm to estimate the size of a study to obtain generalizable results;
4. we analyze two recent experimental studies, Matteucci et al. (2023); Srivastava et al. (2023), and show how well their results generalize.
5. we publish the GENEXPY[1] Python module to repeat our analysis in other studies.

Paper outline: Section 2 discusses the related work, Section 3 formalizes experimental studies, Section 4 defines generalizability and provides the algorithm to estimate the required size of a study for generalizability, Section 5 contains the case studies, and Section 6 describes the limitations and concludes.

## 2 RELATED WORK

We first discuss the literature related to the problem we are tackling, i.e., why experimental studies may not generalize. Second, we overview the existing concept of model replicability, closely related to our work. Finally, we show other meanings that these words can assume in other domains.

**Non-generalizable results.**   It is well known that experimental results can significantly vary based on design choices (Lu et al., 2023; Matteucci et al., 2023; Qin et al., 2023; McElfresh et al., 2022). Possible reasons include an insufficient number of datasets (Dehghani et al., 2021; Matteucci et al., 2023; Alvarez et al., 2022; Boulesteix et al., 2015) as well as differences in hyperparameter tuning (Bouthillier et al., 2021; Matteucci et al., 2023), initialization seed (Gundersen et al., 2023), and hardware (Zhuang et al., 2022). As a result, the statistical benchmarking literature advocates for experimenters to motivate their design choices (Bartz-Beielstein et al., 2020; Mechelen et al., 2023; Boulesteix et al., 2017; Bouthillier et al., 2021; Montgomery, 2017) and clearly state the hypotheses they are attempting to test with their study (Bartz-Beielstein et al., 2020; Moran et al., 2023).

**Replicability and generalizability in ML.**   Our work formalizes the definitions of replicability and generalizability given in Pineau et al. (2021) and National Academies of Science, 2019. Intuitively, replicable work consists of repeating an experiment on different data, while generalizable work varies other factors as well—e.g., quality metric, implementation. A recent line of work, initiated by (Impagliazzo et al., 2022), has linked replicability to model stability: a $\rho$-replicable model learns (with probability $1 - \rho$) the same parameters from different i.i.d. samples. This definition has later been adapted and applied to other learning algorithms (Esfandiari et al., 2023a), clustering (Esfandiari et al., 2023b), reinforcement learning (Eaton et al., 2023; Karbasi et al., 2023), convex optimization (Ahn et al., 2022), and learning rules (Kalavasis et al., 2023). Recent efforts have been bridging the gap between replicability, differential privacy, generalization error, and global stability (Bun et al., 2023; Chase et al., 2023; Ghazi et al., 2023; Moran et al., 2023; Dixon et al., 2023). However, these applications remain limited to model replicability.

**External validity.**   The external validity of a study is a well-studied concept in the context of causal inference, its main applications being in the social and political sciences (Campbell, 1957). In general, the external validity of a study performed con cerns whether repeating a study on different samples affects the validity of its findings. Generalizability is an aspect of external validity, where

[1]https://anonymous.4open.science/r/genexpy-B94D

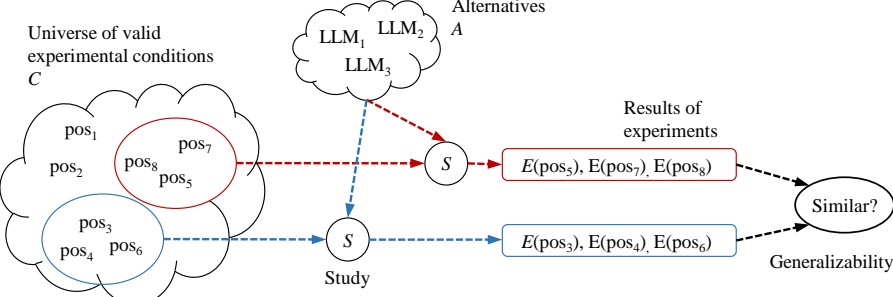

Figure 1: Two empirical studies on the checkmate-in-one task, cf. Example 3.1.

the samples are assumed to come from the same population (Findley et al., 2021). Existing methods assess the sign- and effect-generalization of the treatment on some response variable (Egami & Hartman, 2023). They are thus not applicable to our use-case of ML experimental studies, for which there is—arguably—no treatment and no response variable.

## 3 EXPERIMENTS AND EXPERIMENTAL STUDIES

An *experimental study* is a set of *experiments* comparing the same *alternatives* under different *experimental conditions*. An experimental condition is a tuple of *levels* of *experimental factors*, the parameters defining the experiments. Different factors play different roles in the study: the *design* and *held-constant* factors are fixed by design, while the generalizability of a study is defined in terms of the *allowed-to-vary* factors. The study aims at answering a *research question*, which defines its *scope* and *goals*.

*Example* 3.1. (The "checkmate-in-one" task, cf. Figure 1) An experimenter wants to compare three Large Language Models (LLMs), the *alternatives*, on the "checkmate-in-one" task (Srivastava et al., 2023; Alexander, 2020; Ammanabrolu et al., 2019; 2020; Dambekodi et al., 2020). The assignment is to find the unique checkmating move from a position of pieces on a chessboard: an LLM succeeds if and only if it outputs the correct move. The experimenter considers two *experimental factors*: the number of shots, $m$, and the initial position on the chessboard, $pos_l$. The number of shots is a *design factor*, while the initial position is an *allowed-to-vary* factor. The experimenter wants to find if $LLM_1$ ranks consistently against the other two LLMs when changing the initial position, for a fixed number of shots.

The rest of this section defines the terms introduced above.

### 3.1 EXPERIMENTS

An experiment evaluates all the *alternatives* under a *valid experimental condition*. The *result* of an experiment is a ranking of the alternatives—our choice is detailed and motivated in Appendix A.1.

**Alternatives.** An alternative $a \in A$ is an object compared in the study, like an LLM in Example 3.1. Here, $A$ is the set of alternatives considered in the study, with cardinality $n_a$.

**Experimental factors.** An experimental factor is *anything* that may affect the result of an experiment. $i$ denotes a factor, $C_i$ the (possibly infinite) set of *levels* $i$ can take, $c \in C_i$ a level of $i$, and $I$ the set of all factors. We adapt Montgomery's classification of experimental factors (Montgomery, 2017, Chapter 1) and distinguish between *design*, *held-constant*, and *allowed-to-vary* factors.

- *Design factors* are chosen by the experimenter; e.g., whether and how to tune the hyperparameters, quality metrics, number of shots.
- *Held-constant factors*, e.g., implementation, initialization seed, number of cross-validated folds, may affect the outcome but are not in the scope of the experiment and are fixed by the experimenter.

- *Allowed-to-vary factors*, e.g., "dataset" or "chessboard position" in Example 3.1, may affect the outcome but cannot be held constant: the experimenter expects results to generalize w.r.t. these factors; $I_{\mathrm{atv}}$ denotes them.

**Experimental conditions.** An *experimental condition* $\mathbf{c}$ is a tuple of levels of experimental factors, $\mathbf{c} = (c_i)_{i \in I} \in C \subseteq \prod_{i \in I} C_i$. We endow $C$ with a probability $\mu$, as we will need to sample from it to define the result of a study in Section 3.2. The probability space $(C, \mathcal{F}, \mu)$ is the *universe of valid experimental conditions*. $C$ may not coincide with $\prod_{i \in I} C_i$ as some experimental conditions may be *invalid*, i.e., illegal or not of interest. Validity has to be assessed on a case-by-case basis. For instance, in Example 3.1, $C = \{(\mathrm{pos}_l, m)\}_{l,m}$, where $\mathrm{pos}_l$ is a legal configuration of pieces on a chessboard and $m$ is the non-negative number of shots.

**Definition 3.1** (Rankings with ties). A ranking $r$ on $A$ is a transitive and reflexive binary endorelation on $A$. Equivalently, $r$ is a totally ordered partition of $A$ into *tiers* of equivalent alternatives. $r(a)$ denotes the *rank* of $a \in A$, i.e., the position of the tier of $a$ in the ordering. W.l.o.g. $(\mathcal{R}_{n_a}, \mathcal{P}(\mathcal{R}_{n_a}))$ denotes the measure space of all rankings of $n_a$ objects, where $\mathcal{P}$ indicates the power set.

**Experimental results.** The *experiment function* $E$ evaluates the alternatives $A$ under a valid experimental condition $\mathbf{c} \in C$. Unless necessary, we consider $A$ fixed and omit it in our notation. We require that $E : C \to \mathcal{R}_{n_a}$ is a measurable function, for some fixed $A$. Finally, the *result* of an experiment $E(A, \mathbf{c})$ is a ranking on $A$. We

*Example* 3.1 (Continued). The result of an experiment on $(\mathrm{pos}_l, n)$ is a ranking of the three LLMs, according to whether or not they output the checkmating move. Suppose that only $\mathrm{LLM}_1$ and $\mathrm{LLM}_2$ output the correct move. Then $E(\mathrm{pos}_l, n)$ ranks $\mathrm{LLM}_1$ and $\mathrm{LLM}_2$ tied as best and $\mathrm{LLM}_3$ as worst.

## 3.2 EXPERIMENTAL STUDIES

A study is defined by its *research question* $\mathcal{Q}$, i.e., its *scope* and *goals*. The *scope* consists of the alternatives $A$, the valid experimental conditions $C$, and the allowed-to-vary factors $I_{\mathrm{atv}}$. The *goal* is the kind of conclusions one is attempting to draw from the study. For now, the goal is a statement of interests, i.e., a set of strings.

**Definition 3.2** (Research question). The research question $\mathcal{Q} = (A, C, I_{\mathrm{atv}}, \mathrm{goals})$ is a tuple containing the set of alternatives $A$, the experimental conditions $C$, the set of allowed-to-vary-factors $I_{\mathrm{atv}}$, and the goals of the study.

*Example* 3.1 (Continued). The research question of the "checkmate-in-one" study is as follows. The *scope* is $\left( A = \{\mathrm{LLM}_a\}_{a=1,2,3}, C = \{(\mathrm{pos}_l, n)\}_{l,n}, I_{\mathrm{atv}} = \{\text{"position"}\} \right)$. The *goal* is "Does $\mathrm{LLM}_1$ rank consistently against the other LLMs?"

A crucial element of our formalization is the distinction between *ideal* and *empirical* studies. An ideal study exhausts its research question; however, its result is not observable. An empirical study is an observable sample of an ideal study.

### 3.2.1 IDEAL STUDIES

The *ideal study* on a research question $\mathcal{Q} = (A, C, I_{\mathrm{atv}}, \mathrm{goals})$ is the experimental study consisting of an experiment for each valid experimental condition $\mathbf{c} \in C$. We say that such a study *exhausts* $\mathcal{Q}$. Hence, there exists exactly one ideal study on $\mathcal{Q}$. The *result* of an ideal study is the probability distribution of the results of its experiments. Recall that the experiment function $E : (C, \mathcal{F}, \mu) \to (\mathcal{R}_{n_a}, \mathcal{P}(\mathcal{R}_{n_a}))$ is measurable.

**Definition 3.3** (Result of an ideal study). The *result of an ideal study* with research question $\mathcal{Q} = (A, C, I_{\mathrm{atv}}, \mathrm{goals})$ is

$$S(\mathcal{Q}) = \mathbb{P} : \mathcal{R}_{n_a} \to [0, 1]$$
$$r \mapsto \mathbb{P}(r) := \mu\left(E^{-1}(r)\right),$$

where $E^{-1}(r) = \{\mathbf{c} : E(\mathbf{c}) = r\} \subseteq C$ is the preimage of $r$ through $E$.

In general, multiple experiments of a study may yield identical results. Definition 3.3 supports this by assigning a higher probability mass to results that occur more often.

### 3.2.2 EMPIRICAL STUDIES

Consider again a research question $\mathcal{Q} = (A, C, I_{\text{atv}}, \text{goals})$. In practice, as $C$ might be infinite or too large, one can only run experiments on a sample of $N$ valid experimental conditions $\{\mathbf{c}_j\}_{j=1}^N \overset{\text{iid}}{\sim} (C, \mu)$. The study performed on $\{\mathbf{c}_j\}_{j=1}^N$ is *an empirical study* on $\mathcal{Q}$, of *size* $N$. In what follows, we will always use $N$ to refer to the size of an empirical study. As for ideal studies, the result of an empirical study is the probability distribution of the results of its experiments.

**Definition 3.4** (Result of an empirical study). *The result of an empirical study* on $\mathcal{Q}$ is

$$\hat{S}_N(\mathcal{Q}) : \mathcal{R}_{n_a} \to [0, 1]$$
$$r \mapsto \# \left\{ j \in \{\mathbf{c}_j\}_{j=1}^N : E(A, \mathbf{c}_j) = r \right\}.$$

Where $\mathcal{Q}, \{\mathbf{c}_j\}_{j=1}^N$ is a research question and a set of valid experimental conditions as above.

The result of an empirical study can be thought of as the empirical distribution of a sample following the distribution of the result of the corresponding ideal study. With a slight abuse of notation, indicating both the sample and its empirical distribution as $\hat{S}_N(\mathcal{Q})$, we write

$$\hat{S}_N(\mathcal{Q}) \overset{\text{iid}}{\sim} S(\mathcal{Q}).$$

## 4 GENERALIZABILITY OF EXPERIMENTAL STUDIES

The currently accepted definition of generalizability is the property of two independent studies with the same research question to yield similar results National Academies of Science, 2019 and Pineau et al. (2021). Although intuitive, this notion is not directly applicable as it does not provide a way to measure the generalizability of a study. We now introduce a quantifiable notion of generalizability of experimental studies, as the probability that any two empirical studies approximating the same ideal study yield similar results.

**Definition 4.1** (Generalizability). Let $\mathcal{Q} = (A, C, I_{\text{atv}}, \kappa)$ be the research question of an ideal study, let $\mathbb{P} = S(\mathcal{Q})$ be the result of that study, and let $d$ be some distance between probability distributions. The *generalizability of the ideal study* on $\mathcal{Q}$ is

$$\text{Gen}(\mathcal{Q}; \varepsilon, n) := \mathbb{P}^n \otimes \mathbb{P}^n \left( (X_j, Y_j)_{j=1}^n : d(X, Y) \leq \varepsilon \right),$$

where $\varepsilon \in \mathbb{R}^+$ is a *similarity threshold*.

As the result of an ideal study—$\mathbb{P}$—is usually unobservable (cf. Section 3.2), we rely on the result of an empirical study, $\hat{\mathbb{P}}_N = \hat{S}_N(\mathcal{Q})$, which approximates $\mathbb{P}$ under the assumption that the experimental conditions are i.i.d. samples from $C$. As the sample size $N$ increases (the empirical study becomes larger), $\hat{\mathbb{P}}_N$ converges in distribution to $\mathbb{P}$.

Definition 4.1 requires a distance $d$ between probability distributions. In the next sections, we propose to use a generalizability based on kernels and the Maximum Mean Discrepancy (MMD) (Gretton et al., 2006), as it allows to capture the goal of a study with an appropriate kernel. We conclude this section with an algorithm to estimate the number of experimental conditions required to obtain generalizable results.

### 4.1 SIMILARITY BETWEEN RANKINGS — KERNELS

Whether two experimental results (i.e., rankings) are similar or not ultimately depends on the goal of the study. For instance, consider two rankings on $A = \{a_1, a_2, a_3\}$, $\mathbf{r} = (1, 2, 3)$ and $\mathbf{r}' = (1, 3, 2)$, where $r_i$ is the tier of alternative $a_i$. The conclusions drawn from $r$ and $r'$ are identical if one's goal is to find the best alternative, but very different if one's goal is to obtain an ordering of the alternatives. One can use kernels to quantify the similarity between experimental results. Kernels are suitable to formalize the aspects of the result of a study one wants to generalize, i.e., the goals of the study. For instance, one kernel is suitable to identify the best tier while another kernel focuses on the position of a specific alternative. In the following, we describe three representative kernels that cover a wide spectrum of possible goals.

**Borda kernel.** The Borda kernel is suitable for goals in the form "Is the alternative $a^*$ consistently ranked the same?". It uses the Borda count: the number of alternatives (weakly) dominated by a given one (Borda, 1781). For a pair of rankings, we compute the Borda counts of $a^*$, and then take their difference.

$$\kappa_b^{a^*,\nu}(r_1, r_2) = e^{-\nu|b_1 - b_2|},$$

where $b_l = \#\{a \in A : r_l(a) \geq r_l(a^*)\}$ is the number of alternatives dominated by $a^*$ in $r_l$ and $\nu \in \mathbb{R}$ is the kernel bandwidth. The Borda kernel takes values in $[e^{(-\nu n_a)}, 1]$. If $\nu$ is too large compared to $1/|b_1 - b_2|$, the kernel is oversensitive and will penalize every deviation too much. On the contrary, if $\nu$ is too small, the kernel is undersensitive and will not penalize deviations unless they are very large. As $|b_1 - b_2| \in [0, n_a]$, we recommend $\nu = 1/n_a$.

**Jaccard kernel.** The Jaccard kernel is suitable for goals in the form "Are the best alternatives consistently the same ones?". As it measures the similarity between sets (Gärtner et al., 2006; Bouchard et al., 2013), we use it to compare the top-$k$ tiers of two rankings.

$$\kappa_j^k(r_1, r_2) = \frac{\left|r_1^{-1}([k]) \cap r_2^{-1}([k])\right|}{\left|r_1^{-1}([k]) \cup r_2^{-1}([k])\right|},$$

where $r^{-1}([k]) = \{a \in A : r_1(a) \leq k\}$ is the set of alternatives whose rank is better than or equal to $k$. The Jaccard kernel takes values in $[0, 1]$.

**Mallows kernel.** The Mallows kernel is suitable for goals in the form "Are the alternatives ranked consistently?". It measures the overall similarity between rankings (Jiao & Vert, 2018; Mania et al., 2018; Mallows, 1957). We adapt the original definition in (Mallows, 1957) for ties,

$$\kappa_m^\nu(r_1, r_2) = e^{-\nu n_d},$$

where $n_d = \sum_{a_1, a_2 \in A} |\text{sign}(r_1(a_1) - r_1(a_2)) - \text{sign}(r_2(a_1) - r_2(a_2))|$ is the number of discordant pairs and $\nu \in \mathbb{R}$ is the kernel bandwidth. If a pair is tied in one ranking but not in the other, one counts it as half a discordant pair. The Mallows kernel takes values in $\left[\exp\left(-2\nu\binom{n_a}{2}\right), 1\right]$. If $\nu$ is too large compared to $1/n_d$, the kernel is oversensitive and it will penalize every deviation too much. On the contrary, if $\nu$ is too small, the kernel is undersensitive and will not penalize deviations unless they are very large. As $n_d \in \left[0, \binom{n_a}{2}\right]$, we recommend $\nu = 1/\binom{n_a}{2}$.

## 4.2 Distance between distributions — Maximum Mean Discrepancy

Having sorted out how to measure the similarity between the results of experiments, we now discuss how to measure the distance between the results of studies. We chose the maximum mean discrepancy (MMD) (Gretton et al., 2006), for the following reasons. First, the MMD takes into consideration the goal of a study, as it requires a kernel—such as the ones described in Section 4.1. Second, it handles sparse distributions well; this is needed as empirical studies are typically small compared to the number of all possible rankings, which grows super-exponentially in the number of alternatives. [2] Finally, it comes with bounds and theoretical guarantees, which we will use in Section 4.3.

**Definition 4.2** (MMD (empirical distributions)). Let $X$ be a set with a kernel $\kappa$, and let $\mathbb{Q}_1$ and $\mathbb{Q}_2$ be two probability distributions on $\mathcal{R}_{n_a}$. Let $\mathbf{x} = (x_i)_{i=1}^n$, $\mathbf{y} = (y_i)_{i=1}^m$ be two i.i.d. samples from $\mathbb{Q}_1$ and $\mathbb{Q}_2$ respectively. Then,

$$\text{MMD}(\mathbf{x}, \mathbf{y})^2 := \frac{1}{n^2} \sum_{i,j=1}^n \kappa(x_i, x_j) + \frac{1}{m^2} \sum_{i,j=1}^m \kappa(y_i, y_j) - \frac{2}{mn} \sum_{\substack{i=1\ldots n \\ j=1\ldots m}} \kappa(x_i, y_j).$$

**Proposition 4.1.** *The MMD takes values in $\left[0, \sqrt{2 \cdot (\kappa_{sup} - \kappa_{inf})}\right]$, where $\kappa_{sup} = \sup_{x,y \in X} \kappa(x, y)$ and $\kappa_{inf} = \inf_{x,y \in X} \kappa(x, y)$.*

---

[2]Fubini or ordered Bell numbers, OEIS sequence A000670.

### 4.3 How many experiments ensure generalizability?

When designing a study, an experimenter has to decide how many experiments to run in order to obtain generalizable results. In other words, they need to choose a (minimum) sample size $n^*$ that achieves the desired generalizability $\alpha^*$ and the desired similarity $\varepsilon^*$.

$$n^* = \min\left\{n \in \mathbb{N}_0 : \mathrm{Gen}\left(\mathbb{P}; \varepsilon^*, n\right) \geq \alpha^*\right\}. \tag{1}$$

To estimate $n^*$ we make use of a linear dependency between the logarithms of the sample size $n$ and the logarithm of the $\alpha^*$-quantile of the MMD $\varepsilon_n^{\alpha^*}$ that we have observed in our experiments.

**Proposition 4.2.** $\forall \alpha^*$, *there exist* $\beta_0 \geq 0$ *and* $\beta_1 \leq 0$ *s.t.*

$$\log(n) \approx \beta_1 \log\left(\varepsilon_n^{\alpha^*}\right) + \beta_0 \tag{2}$$

Appendix B.3.2 provides a proof for a simplified case. Proposition 4.2 suggests that one can use a small set of $N$ preliminary experiments to estimate $n^*$. One can then iteratively improve that estimate with the results of additional experiments.

Our algorithm, shown in detail in Appendix B.3.3, requires specifying the desired generalizability, $\alpha^*$, and the similarity threshold between the studies results, $\varepsilon^*$. Then, it performs the following steps:

1. it estimates the $\alpha^*$-quantile of the MMD for all $n$ less than some budget $n_{\max}$. If there exists an $n$ less than $n_{\max}$ that satisfies the condition in (1), we return it as $n^*$;
2. it then fits the linear model in (2), computing the coefficients $\beta_0$ and $\beta_1$;
3. finally, it outputs $n^* = \exp\left(\beta_1 \log\left(\varepsilon_n^{\alpha^*}\right) + \beta_0\right)$, which satisfies the condition in (1) thanks to Proposition 4.2.

In practice, choosing $\varepsilon^*$ is hardly interpretable as it is a threshold on the MMD. To solve this, we propose choosing $\varepsilon^*$ as a function of another parameter $\delta^*$, such that

$$\varepsilon^*(\delta^*) = \sqrt{2(\kappa_{\sup} - f_\kappa(\delta^*))}.$$

Here, $\delta^*$ represents the distance between two rankings as computed by the kernel and $f_\kappa$ is the function linking the distance to the kernel value. For instance, for the Jaccard kernel, $\delta^*$ is simply the Jaccard coefficient between the top-$k$ tiers of two rankings, $f_\kappa(\delta^*) = 1 - \delta^*$, and $\varepsilon^*(\delta^*) = \sqrt{2(1 - (1 - \delta^*))}$. For the Mallows kernel (with our recommendation for $\nu$), $\delta^*$ is the fraction of discordant pairs, $f_\kappa(x) = e^{-x}$, and $\varepsilon^*(\delta^*) = \sqrt{2(1 - e^{-\delta^*})}$. As a concrete example, achieving $(\alpha^* = 0.99, \delta^* = 0.05)$-generalizable results for the Jaccard kernel means that, with probability $0.99$, the average Jaccard coefficient between two rankings drawn from the results is $0.95$.

## 5 Case studies

### 5.1 Case Study 1: A benchmark of categorical encoders

We now evaluate the generalizability of a recent study (Matteucci et al., 2023) that analyzes the performance of encoders for categorical data. The performance of an encoder is approximated by the quality of a model trained on the encoded data. The *design factors* are the model, the tuning strategy for the pipeline, and the quality metric for the model, while the only *allowed-to-vary factor* is the dataset. We impute missing values in the results of the study by assigning the worst rank. We evaluate how well the results of the study generalize w.r.t. three goals:

$(g_1)$ Find out if the one-hot encoder (a popular encoder) ranks consistently amongst its competitors, using the Borda kernel with $\nu = 1/n_a$.
$(g_2)$ Investigate if some encoders outperform all the others using the Jaccard kernel with $k = 1$.
$(g_3)$ Evaluate whether the encoders are typically ranked in a similar order, using the Mallows kernel with $\nu = 1/\binom{n_a}{2}$.

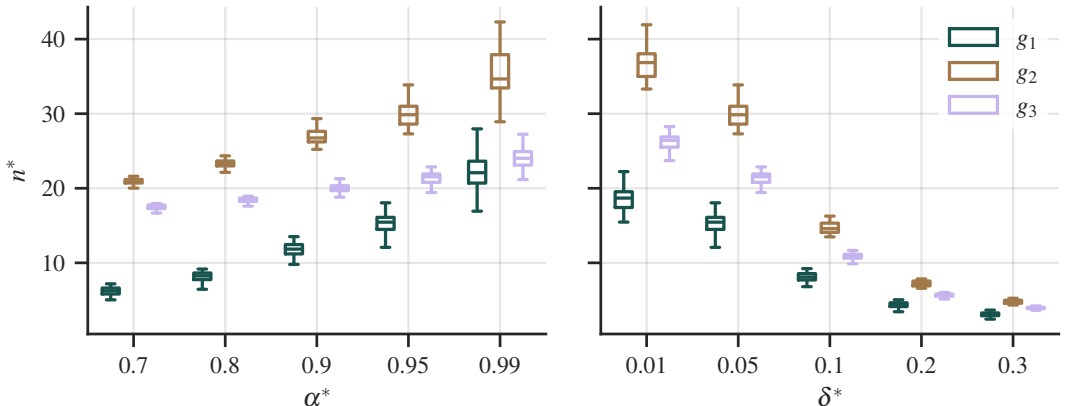

Figure 2: Number of necessary experiments $n^*$ to achieve generalizability for categorical encoders, for different desired generalizability $\alpha^*$, similarity threshold $\delta^*$, goals $g_i$. The variation in the plot is due to the combinations of design factors.

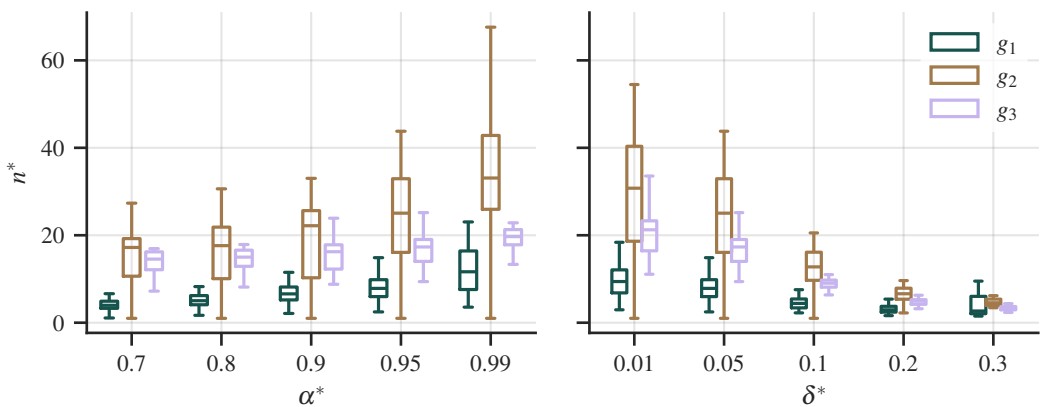

Figure 3: Number of necessary experiments $n^*$ to achieve generalizability for LLMs, for different desired generalizability $\alpha^*$, similarity threshold $\delta^*$, goals $g_i$. The variation in the plot is due to the combinations of design factors.

Figure 2 shows the predicted $n^*$ for different choices of $\alpha^*$ and $\delta^*$, the other one fixed at $0.95$ and $0.05$ respectively. The variance in the boxes comes from variance in the design factors. For example, the results for the design factors "decision tree, full tuning, accuracy" have a different $(\alpha^*, \delta^*)$-generalizability than the results for "SVM, no tuning, accuracy". We observe on the left that—as expected—obtaining generalizable results requires more experiments as the desired generalizability $\alpha^*$ increases. We can also see that the variance of the boxes increases with $\alpha^*$. This means that the choice of the design factors has a larger influence on the achieved generalizability. We observe the same when decreasing $\delta^*$, as it corresponds to a stricter similarity condition on the rankings. In the rather extreme cases of $\alpha^* = 0.7$ or $\delta^* = 0.3$, even less than 10 datasets are enough to achieve $(\alpha^*, \delta^*)$-generalizability.

Consider now goal $g_2$ for two different choices of design factors: (A): "decision tree, full tuning, accuracy" and (B): "SVM, full tuning, balanced accuracy". Furthermore, let $(\alpha^*, \delta^*) = (0.95, 0.05)$: we estimate $n^* = 28$ for (A) and $n^* = 34$ for (B), corresponding to the bottom and top whiskers of the corresponding box in Figure 2. As both (A) and (B) were evaluated using $n = 30$ experiments, we conclude that the results of (A) are (barely) $(0.95, 0.05)$-generalizable, while those of (B) are not. Hence, one should run more experiments with fixed factors (B) to make the study generalizable.

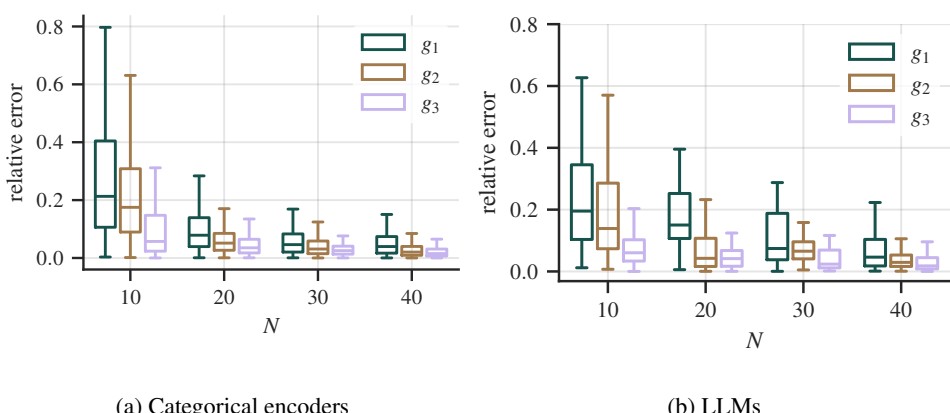

(a) Categorical encoders

(b) LLMs

Figure 4: Relative error between the estimate of $n^*$ from $N$ preliminary experiments and $n_{50}^*$.

## 5.2 CASE STUDY 2: BIG-BENCH — A BENCHMARK OF LARGE LANGUAGE MODELS

We now evaluate the generalizability of BIG-bench (Srivastava et al., 2023), a collaborative benchmark of Large Language Models (LLMs). The benchmark compares LLMs on different tasks, such as the checkmate-in-one task (cf. Example 3.1), and for different numbers of shots. Task and number of shots are the *design factors*. Every task has a number of subtasks, which is the *allowed-to-vary factor*. We stick to the preferred scoring for each subtask. As the results have too many missing values to impute them, we only consider the experimental conditions where at least $80\%$ of the LLMs had results, and to the LLMs whose results cover at least $80\%$ of the conditions.

Similar to before, we define the three goals as follows:

- ($g_1$) Find out if GPT3 (to date, one of the most popular LLMs) ranks consistently amongst its competitors, using the Borda kernel with $\nu = 1/n_a$.
- ($g_2$) Investigate if some encoders outperform all the others using the Jaccard kernel with $k = 1$.
- ($g_3$) Evaluate whether the LLMs are typically ranked in a similar order, using the Mallows kernel with $\nu = 1/\binom{n_a}{2}$.

Figure 3 shows the predicted $n^*$ for different choices of $\alpha^*$ and $\delta^*$, the other one fixed at $0.95$ and $0.05$ respectively. Again, the variance in the boxes comes from variance in the design factors, i.e., the task and the number of shots. As before, increasing $\alpha^*$ or decreasing $\delta^*$ leads to higher $n^*$. Unlike in the previous section, $n^*$ for $g_2$ greatly depends on the combination of fixed factors, as we now detail.

Consider now goal $g_2$ for two different choices of design factors: (A): "conlang_translation, 0 shots", and (B): "arithmetic, 2 shots". Furthermore, let $(\alpha^*, \delta^*) = (0.95, 0.05)$. For this choice of parameters, we estimate $n^* = 44$ for (A), corresponding to the top whisker of the corresponding box in Figure 2. As the study evaluates (A) on 10 subtasks, it is therefore not $(0.95, 0.05)$-generalizable. In fact, we estimate that this would require 34 more subtasks. For (B), on the other hand, we estimate $n^* = 1$: the best 2-shot LLM for the observed subtasks is always PALM 535B. Hence, the result of a single experiment is enough to achieve $(0.95, 0.05)$-generalizability.

Note that, although we correctly estimated $n^* = 1$ for (B), this estimate relies on 10 preliminary experiments. In other words, our algorithm was able to quantify *in hindsight* that a single experiment would have been enough to obtain generalizable results. Of course, however, one cannot trust an estimate of $n^*$ based on only one experiment. The next section thus investigates how the number of preliminary experiments influences the estimate of $n^*$.

## 5.3 HOW MANY PRELIMINARY EXPERIMENTS?

This section evaluates the influence of the number of preliminary experiments $N$ on $n^*$. We consider, for both studies, the design factor combinations for which we have at least $50$ experiments. This results in 23 out of 48 combinations for the categorical encoders and 9 out of 24 combinations for

the LLMs. For each of those combinations, we consider the estimate $n_{50}^*$ made at $N = 50$ as the ground truth and observe how the estimates of $n^*$ for $N < 50$ differ. Figure 4 shows the absolute relative error $|n_N^* - n_{50}^*|/n_{50}^*$, for different goals: the relative errors behave very differently. For goal $g_3$ (Mallows kernel), even $n_{10}^*$ is close to $n_{50}^*$ for a majority of the design factor combinations. On the contrary, one needs 20 to 30 preliminary experiments for goal $g_1$ (Borda kernel). This means that knowing the goals of a study when performing preliminary experiments can help understand how trustworthy the estimate of $n^*$ is.

Appendix C.1 complements this section analyzing the behavior of $n_N^*$ on synthetic data, for which the true $n^*$ is known.

## 6 Conclusion

**Limitations.** First, we modeled experimental results as rankings, their similarity with kernels, and the similarity between distributions of results with the MMD. There are, of course, other possibilities, such as using the raw performance for the experimental results. Second, in Section 5, we post-processed missing evaluations by dropping or imputing them. One could achieve the same by adapting the kernels to missing values.

**Future work.** First, as generalizability only deals with a fixed scope and alternatives, one can include transportability—how well results hold when the scope changes—in our framework. Second, we estimate the distribution of the MMD by sampling multiple times from the results. A non-asymptotic theory of the MMD could speed up this procedure significantly. Third, we plan to provide guarantees on the convergence of $n_N^*$ to the true value of results needed for generalizability, $n^*$.

**Conclusions.** An experimental study is generalizable if, with high probability, its findings will hold under different experimental conditions, e.g., on unseen datasets. Non-generalizeable studies might be of limited use or even misleading. This study is, to our knowledge, the first to develop a quantifiable notion for the generalizability of experimental studies. To achieve this, we formalize experiments, experimental studies and their results—rankings and distributions over rankings. Our approach allows us to estimate the number of experiments needed to achieve a desired level of generalizability in new experimental studies. We demonstrate its utility showing generalizable and non-generalizable results in two recent experimental studies.

## Acknowledgments

...

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

# A  DETAILS FOR SECTION 3

## A.1  WHY RANKINGS?

We chose to formalize experimental results as rankings for the following reasons:

(i) They are already widely used for non-parametric tests such as Friedman, Nemenyi, and Conover-Iman Demsar (2006); Conover & Iman (1982).

(ii) They do not suffer from experimental-condition-fixed effects, such as a dataset being inherently easier to solve than another one. There are multiple ways to deal with these effects, but, there none of these procedures is preferred over the others. A closely related problem is that of consensus ranking aggregation Matteucci et al. (2023); Nießl et al. (2022).

(iii) By defining appropriate kernels for rankings (Section 4.1), we were able to model different goals of a study.

### A.1.1  OTHER POSSIBILITIES

Our framework, relying on the MMD and kernels to compare the results of studies, does not require the results to be rankings. Instead, one can model the experimental results to be elements of an arbitrary probability space $X$, provided that **1.** one can define a kernel on $X$, and **2.** the kernel models the goals of the study. For instance, one can use the raw performance of the algorithms as the result and the Gaussian kernel to compare them. In this case, however, it is unclear what the goal of the corresponding study would be—how to interpret the kernel.

# B  DETAILS FOR SECTION 4

## B.1  DETAILS FOR SECTION 4.1

This section contains the proofs to show that the similarities introduced in Section 4.1 are kernels, i.e., symmetric and positive definite functions. As symmetry is a clear property of all of them, we only discuss their positive definiteness. Our proofs for the Borda and Mallows kernels follow that in (Jiao & Vert, 2018): we define a distance $d$ on the set of rankings $\mathcal{R}_{n_a}$ and show that $(\mathcal{R}_{n_a}, d)$ is isometric to an $L_2$ space. This ensures that $d$ is a conditionally positive definite (c.p.d.) function and, thus, that $e^{-\nu d}$ is positive definite (Schoenberg, 1938; Schölkopf, 2000). Our proof for the Jaccard kernel, instead, follows without much effort from previous results. For ease of reading, we restate the definitions as well.

**Definition B.1** (Borda kernel).
$$\kappa_b^{a^*,\nu}(r_1, r_2) = e^{-\nu|b_1 - b_2|}, \tag{3}$$
where $b_l = \#\{a \in A : r_l(a) \geq r_l(a^*)\}$ is the number of alternatives dominated by $a^*$ in $r_l$ and $\nu \in \mathbb{R}$.

**Proposition B.1.** *The Borda kernel as defined in* (3) *is a kernel.*

*Proof.* Define a distance
$$d : \mathcal{R}_{n_a} \times \mathcal{R}_{n_a} \to \mathbb{R}^+$$
$$(r_1, r_2) \mapsto |b_1, b_2|,$$
where $b_l = \{a \in A : r_l(a) \geq r_l(a^*)\}$ is the number of alternatives dominated by $a^*$ in $r_l$. Now, $(\mathcal{R}_{n_a}, d)$ is isometric to $(\mathbb{R}, \|\cdot\|_2)$ via the map $r_l \mapsto b_l$. Hence, $d$ is c.p.d. and $\kappa_b$ is a kernel.  $\square$

**Definition B.2** (Jaccard kernel).
$$\kappa_j^k(r_1, r_2) = \frac{\left|r_1^{-1}([k]) \cap r_2^{-1}([k])\right|}{\left|r_1^{-1}([k]) \cup r_2^{-1}([k])\right|}, \tag{4}$$
where $r^{-1}([k]) = \{a \in A : r_1(a) \leq k\}$ is the set of alternatives whose rank is better than or equal to $k$.

**Proposition B.2.** *The Jaccard kernel as defined in* (4) *is a kernel.*

*Proof.* It is already know that the Jaccard coefficients for sets is a kernel (Gärtner et al., 2006; Bouchard et al., 2013). As the Jaccard kernel for rankings is equivalent to the Jaccard coefficient for the $k$-best tiers of said rankings, the former is also a kernel. $\square$

**Definition B.3** (Mallows kernel).
$$\kappa_m^\nu(r_1, r_2) = e^{-\nu n_d}, \tag{5}$$
where $n_d = \sum_{a_1, a_2 \in A} |\text{sign}(r_1(a_1) - r_1(a_2)) - \text{sign}(r_2(a_1) - r_2(a_2))|$ is the number of discordant pairs and $\nu \in \mathbb{R}$ is the kernel bandwidth.

**Proposition B.3.** *The Mallows kernel as defined in* (5) *is a kernel.*

*Proof.* The number of discordant pairs $n_d$ is a distance on $\mathcal{R}_{n_a}$ (Snell & Kemeny, 1962). Consider now the mapping of a ranking into its adjacency matrix,

$$\Phi : \mathcal{R}_{n_a} \to \{0, 1\}^{n_a \times n_a}$$
$$r \mapsto (\text{sign}(r(i) - r(j)))_{i,j=1}^{n_a}.$$

Then,
$$n_d = \|\Phi(r_1) - \Phi(r_2)\|_1 = \|\Phi(r_1) - \Phi(r_2)\|_2^2$$
where $\|\cdot\|_p$ indicates the entry-wise matrix $p$-norm and the equality holds because the entries of the matrices are either 0 or 1. As a consequence, $(\mathcal{R}_{n_a}, n_d)$ is isometric to $(\mathbb{R}^{n_a \times n_a}, \|\cdot\|_2)$ via $\Phi$. Hence, $n_d$ is c.p.d. and $\kappa_m$ is a kernel. $\square$

## B.2 DETAILS FOR SECTION 4.2

**Proposition 4.1.** *The MMD takes values in* $\left[0, \sqrt{2 \cdot (\kappa_{sup} - \kappa_{inf})}\right]$, *where* $\kappa_{sup} = \sup_{x,y \in X} \kappa(x, y)$ *and* $\kappa_{inf} = \inf_{x,y \in X} \kappa(x, y)$.

*Proof.*

$$0 \le \text{MMD}_\kappa(\mathbf{x}, \mathbf{y})^2 = \frac{1}{n^2} \sum_{i,j=1}^n \kappa(x_i, x_j) + \frac{1}{m^2} \sum_{i,j=1}^m \kappa(y_i, y_j) - \frac{2}{mn} \sum_{\substack{i=1...n \\ j=1...m}} \kappa(x_i, y_j) \tag{6}$$

$$\le \frac{1}{n^2} \sum_{i,j=1}^n \kappa_{\text{sup}} + \frac{1}{m^2} \sum_{i,j=1}^n \kappa_{\text{sup}} - \frac{2}{mn} \sum_{\substack{i=1...n \\ j=1...m}} \kappa_{\text{inf}}$$

$$= 2(\kappa_{\text{sup}} - \kappa_{\text{inf}})$$

$\square$

## B.3 DETAILS FOR SECTION 4.3

### B.3.1 CHOICE OF $\alpha^*$, $\varepsilon^*$, AND $\delta^*$

Consider a research question $\mathcal{Q} = (A, C, I_{\text{atv}}, \kappa)$ and the corresponding ideal study with result $\mathbb{P}$. The algorithm introduced in Section 4.3 aims at finding the minimum $n^*$ such that, given two independent empirical studies on $\mathcal{Q}$, they achieve similar results. It has two hyperparameters, $\alpha^*$ and $\varepsilon^*$. $\alpha^* \in [0, 1]$ is the generalizability that one wants to achieve from the study, i.e., the probability that two independent realizations of the same ideal study will yield similar results. $\varepsilon^* \in \mathbb{R}^+$ is a similarity threshold: the results of two empirical studies $\mathbf{x}, \mathbf{y} \overset{\text{iid}}{\sim} \mathbb{P}$ are similar if $\text{MMD}_\kappa(\mathbf{x}, \mathbf{y}) \le \varepsilon^*$. However, as it is, $\varepsilon^*$ is not interpretable. Instead, adapting the proof of Proposition 4.1, we can bound the MMD by imposing a condition on the kernel, as we'll now illustrate. The key remark is that we are looking for a condition in the form

$$\text{MMD}_\kappa(\mathbf{x}, \mathbf{y}) \le \varepsilon^* = \sqrt{2(\kappa_{\text{sup}} - \delta')},$$

where $\delta' \in [0, \kappa_{\text{sup}}]$ replaces the third summatory in (6). In other terms, we can interpret $\delta'$ as the minimum acceptable value for the average of the kernel, $\mathbb{E}_{\mathbb{P}^2}[\kappa(x, y)]$. We now go a step further and

compute $\delta'$ (a condition on the kernel) from $\delta^* \in [0,1]$ (a condition on the rankings). The relation between $\delta'$ and $\delta^*$ changes with the kernel, and so does the interpretation of $\delta^*$. For the three kernels we discuss in Section 4.1:

- *Mallows kernel with $\nu = 1/\binom{n}{2}$:* $\delta^*$ is the fraction of discordant pairs, $\delta' = e^{-\delta^*}$.
- *Jaccard kernel:* $\delta^*$ is the intersection over union of the top $k$ tiers, $\delta' = 1 - \delta^*$.
- *Borda kernel with $\nu = 1/n_a$:* $\delta^*$ is the difference in relative position of $a^*$ in the rankings, normalized to the length of the rankings, $\delta' = e^{-\delta^*}$

### B.3.2 PROOF OF PROPOSITION 4.2

**Proposition 4.2.** $\forall \alpha^*$, *there exist $\beta_0 \geq 0$ and $\beta_1 \leq 0$ s.t.*

$$\log(n) \approx \beta_1 \log\left(\varepsilon_n^{\alpha^*}\right) + \beta_0 \tag{2}$$

*Proof.* We provide a proof replacing the sample MMD with the distribution-free bound defined in (Gretton et al., 2012).

$$\mathbb{P}^n \otimes \mathbb{P}^n \left( (X_j, Y_j)_{j=1}^n : \mathrm{MMD}(X,Y) - \left(\frac{2\kappa_{\sup}}{n}\right) > \varepsilon \right) < \exp\left(-\frac{n\varepsilon^2}{4\kappa_{\sup}}\right)$$

$$\overset{(1)}{\Longrightarrow} \mathbb{P}^n \otimes \mathbb{P}^n \left( (X_j, Y_j)_{j=1}^n : \mathrm{MMD}(X,Y) > \varepsilon' \right) < \exp\left(-\frac{n\left(\varepsilon' - \left(\frac{2\kappa_{\sup}}{n}\right)\right)^2}{4\kappa_{\sup}}\right)$$

$$\overset{(2)}{\Longrightarrow} \mathbb{P}^n \otimes \mathbb{P}^n \left( (X_j, Y_j)_{j=1}^n : \mathrm{MMD}(X,Y) > n^{-\frac{1}{2}}\left(\sqrt{-\log(1-\alpha)\,4\kappa_{\sup}} + \sqrt{2\kappa_{\sup}}\right) \right) < 1 - \alpha$$

$$\overset{(3)}{\Longrightarrow} \mathbb{P}^n \otimes \mathbb{P}^n \left( (X_j, Y_j)_{j=1}^n : \mathrm{MMD}(X,Y) \leq n^{-\frac{1}{2}}\left(\sqrt{-\log(1-\alpha)\,4\kappa_{\sup}} + \sqrt{2\kappa_{\sup}}\right) \right) \geq \alpha$$

where:

(1) $\varepsilon' = \varepsilon + \sqrt{2\kappa_{\sup}/n}$.

(2) $1 - \alpha = \exp\left(-\frac{n\left(\varepsilon' - \left(\frac{2\kappa_{\sup}}{n}\right)\right)^2}{4\kappa_{\sup}}\right)$ and $\varepsilon' = n^{-\frac{1}{2}}\left(\sqrt{-\log(1-\alpha)\,4\kappa_{\sup}} + \sqrt{2\kappa_{\sup}}\right)$.

(3) Take the complementary event.

Now,

$$q_n^\alpha = n^{-\frac{1}{2}}\left(\sqrt{-\log(1-\alpha)\,4\kappa_{\sup}}\right) + \sqrt{2\kappa_{\sup}}$$

$$\Rightarrow n = (q_n^\alpha)^{-2}\left(\sqrt{-4\kappa_{\sup}\log(1-\alpha)} + \sqrt{2\kappa_{\sup}}\right)^2$$

$$\Rightarrow \log(n) = -2\log(q_n^\alpha) + 2\log\left(\sqrt{-4\kappa_{\sup}\log(1-\alpha)} + \sqrt{2\kappa_{\sup}}\right).$$

concluding the proof. $\square$

*Remark.* Although theoretically sound, using the abovementioned bound instead of the sample MMD leads to excessively conservative estimates for $n^*$, roughly one order of magnitude greater than the empirical estimate.

### B.3.3 PSEUDOCODE FOR THE ALGORITHM

---

**Algorithm 1** Compute $n_N^*$ from preliminary study

---

**Require:** $\alpha^*$                                                     ▷ desired generalizability
**Require:** $\delta^*$                                           ▷ similarity threshold on rankings
**Require:** $\mathcal{Q}$                               ▷ research question, $\mathcal{Q} = (A, C, I_{\text{atv}}, \kappa)$
**Require:** $N$                                          ▷ size of preliminary study
**Require:** $n_{\max}$                           ▷ maximum sample size to compute the MMD
**Require:** $n_{\text{rep}}$                         ▷ number of repetitions to compute the MMD

    **procedure** ESTIMATENSTAR($\alpha^*, \delta^*, \mathcal{Q}, N, n_{\max}, n_{\text{rep}}$)
        $\varepsilon^* \leftarrow$ compute $\varepsilon^*$ from $\delta^*$                                ▷ cf. Appendix B.3
        sample $\{\mathbf{c}_j\}_{j=1}^N \overset{\text{iid}}{\sim} C$
        $n_{\max} \leftarrow \min \{n_{\max}, [N/2]\}$        ▷ we need two disjoint samples of size $n_{\max}$ from $\{\mathbf{c}_j\}_{j=1}^N$
        **for** $n = 1 \ldots n_{\max}$ **do**
            mmds $\leftarrow$ empty list
            **for** $n = 1 \ldots n_{\text{rep}}$ **do**
                sample without replacement $(\mathbf{c}_j)_{j=1}^{2n_{\max}} \sim \{\mathbf{c}_j\}_{j=1}^N$
                $\mathbf{x} \leftarrow (\mathbf{c}_j)_{j=1}^{n_{\max}}$                            ▷ split the disjoint samples
                $\mathbf{y} \leftarrow (\mathbf{c}_j)_{j=n_{\max}}^{2n_{\max}}$
                append MMD $(\mathbf{x}, \mathbf{y})$ to mmds
            **end for**
            $\varepsilon_n^{\alpha^*} \leftarrow \alpha^*$-quantile of mmds
        **end for**
        fit a linear regression $\log(n) = \beta_1 \log\left(\varepsilon_n^{\alpha^*}\right) + \beta_0$
        $n_N^* \leftarrow \beta_1 \log(\varepsilon^*) + \beta_0$
        **return** $n_N^*$
    **end procedure**

    **procedure** RUNEXPERIMENTS($\alpha^*, \delta^*, \mathcal{Q}, n_{\max}, n_{\text{rep}}, \text{step}$)
        $N \leftarrow$ step
        **while** $n^* > N$ **do**
            sample $\{\mathbf{c}_j\}_{j=1}^N \overset{\text{iid}}{\sim} C$
            $n^* \leftarrow$ ESTIMATENSTAR($\alpha^*, \delta^*, \mathcal{Q}, N, n_{\max}, n_{\text{rep}}$)
            $N \leftarrow N + \text{step}$
        **end while**
    **end procedure**

---

## C  DETAILS FOR SECTION 5

### C.1  PREDICTION OF $n^*$

This section investigates how well our method described in Section 4.3 can predict the correct number of experiments required to ensure generalizability, $n^*$. Recall that, for a desired generalizability $\alpha^*$ and a desired threshold $\varepsilon^*$ obtained as in Appendix B.3,

$$n^* = \min \{n \in \mathbb{N}_0 : \text{Gen}\left(\mathbb{P}; \varepsilon^*, n\right) \geq \alpha^*\} .$$

To do so, we run the following simulation:

1. Uniformly generate 1000 rankings of 5 alternatives, these form the universe $U$.
2. Compute the generalizability of the sample for increasing $n$, and get $n^*$ satisfying C.1.
3. For $N = 10, 20, 40, 80$:
       (a) Sample with replacement $N$ rankings from $U$—simulate running $N$ preliminary experiments.

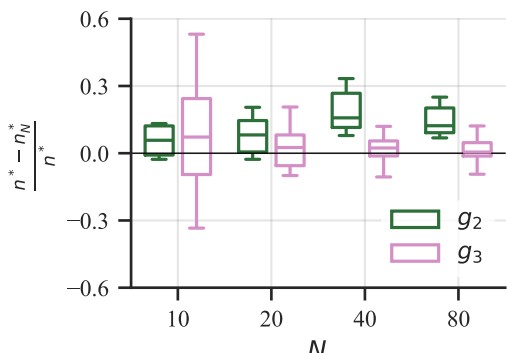

Figure 5: Relative error of the prediction of $n^*$ from $N$ preliminary experiments $(n_N^*)$, for two goals.

     (b) Predict $n_N^*$ from the $N$ preliminary experiments (Section 4.3).
     (c) Compute the relative error $(n^*-n_N^*)/n^*$.

4. Repeat the previous steps 50 times for the Jaccard and Mallows kernels.[3]

The outcome is shown in Figure 5 for the Jaccard $(g_2)$ and Mallows $(g_3)$ kernels. Our method is able. in general, to get within 30% of the correct value of $n^*$ even from 10 preliminary experiments.

---

[3]We did not investigate the Borda kernel as, for synthetic data, there is no clear preferred alternative.

