# OpenReview forum: "Can You Trust Your Experiments? Generalizability of Experimental Studies"
_ICLR.cc/2025/Conference — ICLR 2025 Conference Withdrawn Submission_

### Official Review · Reviewer_rfGB · 2024-11-01

**Soundness:** 2
**Presentation:** 2
**Contribution:** 2
**Rating:** 5
**Confidence:** 3

**Summary:**

This work focuses on the study of generalizability of the experimental results in machine learning researches. It proposes a general framework to formalize experimental studies and their results, provides a quantifiable definition of the generalizability of experimental studies, develops an algorithm to estimate the size of a study to obtain generalizable results, and finally analyzes two recent studies as an example. Additionally, the authors provide an open-source python module for reproducibility of this paper.

The formalization of experimental studies considers almost all factors involved in most machine learning experiments, including alternative algorithms, design factors (e.g., hype-parameters), held-constant factors (e.g., algorithmic implementation), allowed-to-vary factors (e.g., dataset) and the goal of the experiment. Based on the formalization, this paper gives a formal definition of the result of an empirical study. It converts the result of an empirical study to the empirical distribution of a sample following the distribution of the result of the corresponding ideal study. Based on some kernel functions, the authors incorporate traditional maximum mean discrepancy to obtain the minimum sample size, or the number of experiments required, to achieve generalizable experimental results. The authors also conduct experiments on two algorithms to show the effectiveness of the proposed method.

**Strengths:**

This paper is well-written and easy to follow. It focuses on an interesting problem which might be useful for experimental studies of machine learning algorithms, including recent large language models. It has the following strengths:

1. This paper gives a detailed formalization of experimental studies about comparison between machine learning algorithms. This formalization is quite interesting and makes sense in most experiments in machine learning research.

2. The conversion from empirical results to a distribution enables mathematical analysis of the generalization of empirical result, which is important.

3. The authors conduct both theoretical and experimental analysis of the minimum sample size to achieve generalizable experimental results, which verifies the effectiveness of the proposed method.

**Weaknesses:**

1. Although the formalization seems correct and useful, the contributions of this paper are still a bit incremental. The MMD is a very traditional measure between distributions, yet the reason why this paper adopts MMD instead of other measures is not discussed.

2. The threshold of MMD is important since it directly influences the test power. However, in section 4.3, the authors set the threshold as $\sqrt{2(\kappa_{sup}-f_{\kappa}(\delta^*))$ without any theoretical analysis or empirical tests. It would be more convincing if the authors gave more detailed discussion.

3. The case studies in section 5 only focus on two works, and it does not analyze the experimental results in some basic and simple classification tasks, which is more common than the two cases in section 5. This paper would have more significant contributions if more common experimental results were included (e.g., the comparison of different convolution neural networks on image classification / segmentation tasks).

**Questions:**

1. Although the formalization seems correct and useful, the contributions of this paper are still a bit incremental. The MMD is a very traditional measure between distributions, yet the reason why this paper adopts MMD instead of other measures is not discussed.

2. The threshold of MMD is important since it directly influences the test power. However, in section 4.3, the authors set the threshold as $\sqrt{2(\kappa_{sup}-f_{\kappa}(\delta^*))$ without any theoretical analysis or empirical tests. It would be more convincing if the authors gave more detailed discussion.

3. The case studies in section 5 only focus on two works, and it does not analyze the experimental results in some basic and simple classification tasks, which is more common than the two cases in section 5. This paper would have more significant contributions if more common experimental results were included (e.g., the comparison of different convolution neural networks on image classification / segmentation tasks).

---

### Official Review · Reviewer_axsn · 2024-11-04

**Soundness:** 3
**Presentation:** 1
**Contribution:** 2
**Rating:** 6
**Confidence:** 4

**Summary:**

The paper "Can You Trust Your Experiments? Generalizability of Experimental Studies" addresses the challenge of generalizability in machine learning (ML) experimental studies. While reproducibility and significance are well-studied, generalizability—whether results remain consistent under unseen experimental conditions—lacks a quantifiable framework in the ML context. The authors propose a formalization of experimental studies and introduce a quantifiable measure for generalizability using rankings and Maximum Mean Discrepancy (MMD). This framework allows researchers to estimate the minimum number of experiments needed to achieve generalizable results. The authors validate their approach through case studies on categorical encoders and large language models (LLMs), illustrating how well the results generalize and the necessary number of experiments for different conditions. They also release a Python module, GENEXPY, to facilitate further analysis.

**Strengths:**

The paper introduces a well-defined, quantifiable measure for the generalizability of experimental studies, which is both timely and impactful for ML research. This approach is innovative in the context of ML, addressing a significant gap by formalizing generalizability for structured experiments and using MMD to quantify similarity between study outcomes.

The authors provide a thorough formalization of experiments, defining experimental factors, conditions, and the results through a ranking-based approach. They include detailed definitions and mathematical grounding, offering robustness to their proposed framework.

The algorithm provided for estimating the required sample size to achieve generalizability is practical and clearly outlined, allowing researchers to determine how many experiments are necessary to meet desired levels of generalizability. This guidance is valuable for designing studies in various ML domains.

The paper demonstrates its framework on categorical encoders and large language models, highlighting the framework’s versatility and its applicability to different types of ML experiments. The use of multiple kernels to reflect various study goals adds flexibility to the methodology.

**Weaknesses:**

The framework relies on a ranking-based representation of experimental results and specific kernels (Borda, Jaccard, and Mallows) for measuring similarity. This may limit the applicability in studies where results cannot naturally be represented as rankings or where the kernels are not well-suited to capture nuanced differences.

While the case studies are diverse, they do not encompass other common ML setups, such as multi-label classification or time-series prediction. These tasks may involve different experimental conditions that challenge the proposed framework’s generalizability.


 Calculating MMD for large datasets or complex experiments can be computationally intensive, which may hinder scalability. Additionally, estimating the minimum sample size through the proposed iterative approach may be prohibitive in cases requiring a large number of experimental conditions.

Choosing the similarity threshold for generalizability is critical but may be difficult to interpret or set in practice. While the paper provides guidelines, it could benefit from further elaboration.

**Questions:**

How might the framework be adapted for experiments that do not naturally produce ranked outputs? For instance, in cases where raw performance metrics are more informative, could the framework be generalized to use non-ranking-based similarity measures?

 Given that the choice of kernel significantly affects similarity measurement, could you provide guidelines or a decision framework for selecting the appropriate kernel based on study goals?

Have you evaluated the scalability of MMD calculation for studies with large sample sizes or high-dimensional data? Is the GENEXPY tool optimized for such scenarios?

You mention that transportability could be a potential future direction. Could you elaborate on how you envision extending the framework to address transportability (i.e., how well results hold when the experimental scope changes)?

**Details Of Ethics Concerns:**

The paper does not follow the 9 page upper limit for the main paper. I am not sure if it should be the reason for desk rejection, hence, flagging it here.

---

### Official Review · Reviewer_iUji · 2024-11-04

**Soundness:** 3
**Presentation:** 3
**Contribution:** 2
**Rating:** 3
**Confidence:** 4

**Summary:**

The paper proposes a formal framework for analyzing the external validity of experiments in ML. The authors complement their theoretical framework with two case studies.

**Strengths:**

- The paper deals with an important problem (generalization of empirical findings in machine learning beyond the immediate study).
- Overall the paper is well written.
- I appreciate the connection to external validity.
- The case studies are helpful to illustrate the proposed theoretical framework.

**Weaknesses:**

The main limitation I see is the following: when discussing the generalization of empirical findings, the key question often is what other experimental settings the findings apply to. In both case studies the authors provide, they do not discuss this crucial point:

- In Case Study 1, the "allowed-to-vary" factor is the dataset. It would be important to understand what kinds of datasets the empirical findings extend to.

- In Case Study 2, the "allowed-to-vary" factor is the subtask. Similarly to the point above, a key question when analyzing the capabilities of LLMs usually is _what task does the model perform well on_?

Hence overall my impression of the paper is that it sidesteps the key question in the external validity of empirical findings in ML: what makes datasets of tasks similar? When is performance on task or dataset X indicative for performance on task or dataset Y?

So in summary, I find the high-level research direction of external validity very interesting. But unfortunately the submission does not address the core question of what makes experimental settings similar. While I appreciate the careful formal treatment of experimental setups etc. in the paper, my opinion is that this part is less critical in furthering our understanding of external validity in ML experiments.

**Questions:**

- It would be helpful if the authors could describe how their proposed method relates to the bootstrap method for building confidence intervals
- The related work may benefit from discussing overfitting as another example of external validity (or the lack thereof) in ML.
- The difference between "design factors" and "held-constant factors" is not clear.
- Line 180 contains an incomplete sentence ("We").
- Line 239: should the citation here be a `\citep`?

---

### Official Review · Reviewer_LEY9 · 2024-11-06

**Soundness:** 3
**Presentation:** 2
**Contribution:** 3
**Rating:** 5
**Confidence:** 3

**Summary:**

This paper attempts to formalize the notation of generalizability of experimental studies. To do so, the paper starts from the ground up by formalizing alternatives (i.e., baselines/methods in ML experiments), experimental factors (i.e., variables affecting results), and valid experimental conditions (i.e., realizations of factors to consider). A result is defined as a ranking over alternatives (e.g., method 1 is better than method 2). The overall result of a study is a distribution over rankings. A study is considered generalizable if two independent studies give similar results. Because results are distributions, quantifying generalizability can be cast as measuring the distributional distance of two results. This paper uses the MMD (on the space of rankings) as the distributional distance. The paper proposes an algorithm (Sec 4.3) to estimate the number of experiments required to obtain generalizable results. Empirically, to demonstrate practical uses of the algorithm, the paper considers two case studies: 1. generalizability of Matteucci et al., 2023 that analyzes the performance of encoders for categorical data, and 2. generalizability of Srivastava et al., 2023, a collaborative benchmark on LLMs.

**Strengths:**

The attempt to precisely formalize all concepts related to an experiment (i.e., alternatives, experiential factors, results) appears to be original. By seeing the result of a study as a distribution over rankings of alternatives, generalizability can be cast as a question of comparing two distributions. I find this modeling choice interesting and novel. Paper is well structured and mostly well written.  The many definitions that the reader has to go through are made easier by examples along the way (i.e., by connecting to the running example of the “checkmate-in-one” task). These examples are helpful.

**Weaknesses:**

While the paper is precise overall, there are some key parts that are relatively imprecise, making them difficult to understand. For instance, the key concept of generalizability is not precisely defined in Def 4.1 (and not sufficiently discussed). The main algorithm in Sec 4.3 is not fully described and more importantly is not fully justified (particularly from the use of the linear regression model in Eq 2).

**Questions:**

I think this paper offers a unique perspective on concepts related to experiments (i.e., formalizing experiments). I am hopeful that your answers to the following questions will convince me.

**Major questions**

1. Formal concepts are precisely introduced up to the end of Sec 3. In Sec 4, in my opinion, Def 4.1 is arguably the most important piece in this paper. It is unfortunately not precise. What is $n$? I understand that $n$ is not $N$ (i.e., the number of experimental conditions sampled from $C$).

2. Related:  Could you please explain step by step what this expression in Def 4.1 means? $\mathbb{P}^n \otimes \mathbb{P}^n \left( (X_j, Y_j)_{j=1}^n : d(X,Y) \le \varepsilon \right)$. In particular, what is $X_j$ (and hence $Y_j$). By analyzing variable types, I can deduce these are distributions. But distributions of what? Is each $X_j$ a distribution as defined in either Def 3.3 or Def 3.4? Are $X_j$ and $Y_j$ paired? It is unfortunate that I missed this crucial detail. This made me unable to fully understand Sec 4.3.

3. Def. 3.1: Do you allow two different alternatives to have the same rank? For instance, $A= (a_1, a_2, a_3)$. Is $r = (1, 2.5, 2.5)$ a valid ranking?

4. Sec 3.2.2. For each of the $N$ experimental conditions $\mathbf{c}_1, \ldots, \mathbf{c}_N$, do you run only one experiment so that in the end we have $N$ experiments (each corresponding to an experimental condition)?

5. $\mathbf{c}_1, \ldots, \mathbf{c}_N$ are assumed i.i.d. In practice, I suppose we do not pre-sample these experimental conditions in advance. Rather, we likely consider a few conditions. Run some experiments to check their results. We then decide whether more experimental conditions need to be considered. This iterative process will not technically give i.i.d. conditions. How does this affect  the i.i.d. assumption and the results of the proposed algorithm?


6. Def 3.4. Is there a normalizer missing when computing the empirical probability?

7.  The main proposed algorithm in Sec 4.3 should be expanded to contain more explanation and details. Why are the three steps the way they are? What is the motivation? What do you need to fit a linear model in step 2? Crucially, what is the training data to fit this linear model?

8. Sec 5.2, L455 “we only consider the experimental conditions where at least 80% of the LLMs had results”. Presumably this translates to restricting the set $C$ of valid experimental conditions. In practice, how do you distinguish or choose to consider between 1. restricting the set $C$, 2. keeping a large set $C$ but treating this filtering as reducing $N$ (in $\mathbf{c}_1, \ldots, \mathbf{c}_N$)? How does this choice affect the result of the algorithm?


**Minor questions**

1. Sec 3.1. Can $A$ be an infinite set?

**Minor comments**

* Title: “Can your trust your…” should be “Can you trust your…”.

* If I may suggest, I think Fig 1 is better removed to get more space to make the algorithm discussed in Sec 4.3 more precise. In fact, you also still have more space left.

* Def 4.1. $\kappa$ is undefined at that point. Is it “goals” or a kernel?

* Def 4.2. Say that $\kappa$ is defined on $X \times X$.

---

### Note · Authors · 2024-11-26

**Comment:**

We thank all the reviewers for their work and their valuable comments, which we will integrate in a future version of the paper.

**Withdrawal Confirmation:**

I have read and agree with the venue's withdrawal policy on behalf of myself and my co-authors.